# The Prevalence of Depression Symptoms and Their Socioeconomic and Health Predictors in a Local Community with a High Deprivation: A Cross-Sectional Studies

**DOI:** 10.3390/ijerph191811797

**Published:** 2022-09-19

**Authors:** Maciej Polak, Grzegorz Józef Nowicki, Katarzyna Naylor, Robert Piekarski, Barbara Ślusarska

**Affiliations:** 1Department of Epidemiology and Population Studies, Jagiellonian University Medical College, Skawińska 8 Str., PL-31-066 Krakow, Poland; 2Department of Family and Geriatric Nursing, Medical University of Lublin, Staszica 6 Str., PL-20-081 Lublin, Poland; 3Department of Didactics and Medical Simulation, Medical University of Lublin, Chodźki 4 Str., PL-20-093 Lublin, Poland; 4Diabetology with Endocrine—Metabolic Laboratory, Department of Paediatric Endocrinology, Medical University of Lublin, Gębali 6 Str., 20-093 Lublin, Poland

**Keywords:** depression symptoms, PHQ-9, local community, socioeconomic predictors, health predictors

## Abstract

Depression is a heterogeneous and etiologically complex psychiatric syndrome thatshows a strong sexual dimorphism and often impacts people with a low socioeconomic status (SES). The aim of the study was to estimate the occurrence of depression symptoms in a local community with a high deprivation rate, the example being the inhabitants of the JanówLubelski County in eastern Poland. A cross-sectional study was carried out on 3752 people aged between 35 and 64. The prevalence of depression symptoms was assessed using the Patient Health Questionnaire-9 (PHQ-9) scale. In the screening for depression symptoms in the entire population we studied, the risk of depression symptoms was 16.1% (*n* = 605), with women having a significantly higher mean score than men (*p* < 0.001). Significant predictors associated with the achievement of 10 points and more in the PHQ-9 assessment in the case of women and men were: living alone, education and having comorbidities. Moreover, female participants living in rural areas were significantly more likely to exhibit depression symptoms, whereas smoking was a significant predictor of depressive symptoms in men. It was observed that in the case of obese women, the chance of being in the higher category of the PHQ-9 assessment was 1.41 times higher than in women with normal body weight. However, in the case of men, an increase in age by one year increased the chance of being in a higher category by 1.02 times. Moreover, the odds of falling into a higher category, as assessed by the PHQ-9 questionnaire, among men who drink alcohol more than once a week was 1.7 times higher than in men who do not drink or consume alcohol occasionally. Summarising the results of studies conducted in a local community characterised by a high deprivation rate, socioeconomic and health variables related to SES significantly impacted the incidence of depression, but they differ in terms of gender.

## 1. Introduction

Depression is a heterogeneous and etiologically complex psychiatric syndrome, often manifested by anxiety disorders, substance use disorders, ADHD, and other co-morbid psychiatric diseases, shows a strong gender dimorphism [1] and often impacts people with low socioeconomic status (SES) [2]. Depression and depressive symptoms are the most common psychological problems in the world, impacting over 300 million people of all ages (4.4% of the world’s population), with an 18.4% increase in people suffering from depression between 2005 and 2015 [3]. The World Health Organisation (WHO) predicts that depression will be the leading cause of disability and a significant contributor to the global disease burden by 2030 [4]. In 2014, 7.1% of the European Union (EU) population reported suffering from chronic symptoms of depression [5]. Moreover, depression is a major factor associated with suicide, leading to 800,000 deaths annually [4]. The burden of depression is a complex concept with different connotations, encompassing the burden on the patient, caregiver, healthcare system, society and the economy. The costs of depression in Europe amount to EUR 92 billion a year, with a large part of the financial burden caused by the loss of productivity by impacted people [6]. As for Poland, few national studies assess the prevalence of depressive symptoms in the population. In Poland, no systematic epidemiological studies on the prevalence of mental disorders have been conducted so far in the general population. The first such study was the study entitled “Epidemiology of psychiatric disorders and access to psychiatric health care—EZOP Poland”, which was conducted on a representative group of 10,082 people (including 5199 women) aged 18–64, and the research report was published in 2012 [7]. According to the cited studies, the prevalence of major depression with a hierarchy of symptoms was present in 3% of the population and minor depression in 0.4%. Another population study (EZOP II) was conducted in 2021, but its results have not yet been published.

Several studies have assessed the relationship between personal and socioeconomic variables, such as gender, age, marital status, education, income level, employment status and social class, and the occurrence of depressive disorders [4,8,9,10]. The results of epidemiological studies show that personal and socioeconomic factors are associated with differences in the prevalence of depressive disorders. Moreover, the prevalence of these factors may change over different periods of time over the years and is often associated with economic changes or crises [11]. Among the countries of Central and Eastern Europe, Poland was the first country to initiate political changes and the transition to a market economy in 1989 [12]. As a result of these changes, there was a gradual overall improvement in living conditions and quality of life. However, the benefits did not extend evenly to all segments of society [13]. People with higher education were more mobile in the labor market, which resulted in higher income. On the other hand, the decline of heavy industry and lower demand for manual labour led to the deterioration of the economic situation of a large group of workers with a low level of education [13,14]. The worse SES of these groups was associated with the intensification of pre-existing social problems and limited access to healthcare [15]. The growing social disproportions may probably influence the regional differentiation of the occurrence of symptoms of depression in populations.

The frequency and degree of depression in local communities with their specific characteristics are still insufficiently studied, especially in local communities with low SES, which is one of the main predictors of health at every stage of life [16]. It is a complex, multi-dimensional social structure conceptualised to account for income, education, and occupation (and other factors), which are often interrelated. However, each SES measure likely reflects slightly different forces related to health and disease at the individual and social levels [17]. An example of such a local community characterised by a low SES is the inhabitants of the Janów Lubelski County in eastern Poland. Socioeconomic status is expressed on the basis of the phenomenon of deprivation at the county level, calculated on the basis of an index taking into account five areas/dimensions of deprivation: population income, employment, living conditions, education and access to goods and services. The Janów Lubelski County indicated in 2013 that it has 20% of those in the whole of the Lubelskie Voivodeship who are the most threatened by deprivation [18]. Moreover, the analysis of many socioeconomic factors of people living in this area in the period preceding the survey was quite unfavourable compared to the national indicators. For example, as of the day preceding the study, in Janów Lubelski County, there was a higher percentage of people with only primary education in relation to the general population (23.2% vs. 27.55%) [19], and the unemployment rate was 15.6% of people in the productive age compared to an unemployment rate of 14% in the country [20].

Furthermore, the indicator concerning the number of people in families using social assistance for the Janów Lubelski County was relatively high—the social welfare system covered as much as 14.6% of the population (for comparison, the average in the Lubelskie Voivodeship is 9.4%) [21]. Therefore, the aim of the research undertaken was to estimate the occurrence of symptoms of depression in a local community with a high deprivation index using the example of the inhabitants of the Janów Lubelski County in eastern Poland. The second goal was to assess the dependence of selected socioeconomic and health predictors on the occurrence of depression symptoms in gender strata.

## 2. Materials and Methods

### 2.1. Study Design and Participants

A prevention and health promotion program was conducted among the inhabitants of the Janów Lubelski county located in eastern Poland in the Lubelskie Voivodeship, “Take Your Health to Heart” (“Weź sobie zdrowie do serca”), between 14 June 2015, and 20 March 2016. Implementing the program financed under the PL 13 Program “Reducing Social Inequalities in Health” of the Norwegian Financial Mechanism 2009–2014 in Janów Lubelski County was possible because the financing covered local communities with high standardised mortality rates (SMR). Therefore, the list of counties eligible to participate in the competition was developed based on the highest standardised mortality rates (SMR) in 2009–2011 in the following categories: malignant neoplasms, cardiovascular diseases, respiratory diseases, digestive system diseases, external causes and total mortality. The county of Janów was also on such a list in the third position (SMR = 1.357) out of 38 counties with the highest standardised mortality rates due to cardiovascular diseases in Poland [22].

As of the day preceding the study, the population of JanówLubelski County amounted to 47,500 people. The project was dedicated to people between 35 and 64 years of age, with 18,827 living in the JanówLubelski County. Recruitment and research promotion was carried out by county and commune local governments and cooperating institutions (religious associations, workplaces, associations, and organisations of utility). Additionally, by directing telephone invitations to persons from the lists of authorised persons. In order to have equal access to participation in the study, 15 points were organised in the Janów Lubelski County, where respondents could report (14 mobile—itinerant points located in various towns and one stationary point located in the City Hospital, which also had a coordinating function). Four thousand and forty people applied to participate in the study. The overall participation rate in the study was 21.45% of eligible persons. All study participants gave their free and informed consent to participate in the study in a specially prepared form. The study was approved by the Bioethics Committee at the Medical University of Lublin (No. KE-0254/112/2014) and was carried out under the Helsinki Declaration.

Amongst volunteered study participants (*n* = 4040), 288 individuals who reported having a history of a cardiovascular event (heart attack or stroke) or physician-diagnosed ischemic heart disease were excluded. Ultimately, 3752 respondents were included in the analysis (Figure 1). The study excluded people who: (1) had a history of a cardiovascular incident (heart attack or stroke); (2) were diagnosed with ischemic heart disease; (3) pregnant women; (4) the respondent could not give informed consent to participate in the study; (5) a person constantly immobilised in bed; (6) a person in a care home or prison.

### 2.2. Data Collection

The data collection team included specially trained nurses. All study participants completed the questionnaire and underwent anthropometric tests (weight and height measurements).

#### 2.2.1. Depressive Symptoms

Patient Health Questionnaire-9 (PHQ-9) [23] was used to assess the occurrence of depression symptoms in the last two weeks. The questionnaire is a self-report tool and consists of nine questions which respondents indicate on a scale from 0 (“not at all”) to 3 (“almost every day”), the frequency with which they experience the following symptoms: anhedonia, depressed mood, sleep disturbance, fatigue, appetite changes, low self-esteem, concentration problems, psychomotor disturbances, and suicidal ideation. Nine items from the questionnaire are consistent with the nine diagnostic criteria for major depressive episodes according to the Diagnostic and Statistical Manual of Mental Disorders (DSM). The overall scale results are calculated as the sum of the 9 items (range from 0 to 27). PHQ-9 has been recommended by the United States Preventive Services Task Force (USPSTF) and other organisations for screening for depression symptoms in primary care and depression screening in the general population. However, the recommendations do not specify a scoring method to be used [24]. The categories of depression severity were initially defined as none (0–4 points), mild (5–9 points), moderate (10–14 points), moderately severe (15–19 points), and severe (20–27 points) [25,26].

#### 2.2.2. Anthropometric Measurements

Anthropometric measurements of height and body weight were performed on all the subjects. Height was measured to an accuracy of 0.1 cm with an altimeter, and weight without shoes or other clothing was measured with a platform scale with an accuracy of 0.1 kg. BMI was calculated and defined as body weight (kg) divided by height squared in meters (kg/m^2^). The following classification of BMI was adopted: 18.5 and 24.9 kg/m^2^—normal, 25–29.99 kg/m^2^—overweight and ≥30 kg/m^2^—obesity [27].

#### 2.2.3. Other Socioeconomic and Health Variables

Data such as age, sex, place of residence, marital status, education, smoking status, frequency of alcohol consumption, living together with someone in the household and comorbidities (the following chronic diseases diagnosed by a doctor: hypertension or diabetes or hypercholesterolaemia) were collected using a standard interview questionnaire.

Smoking status was defined as: non-smoker (if the respondent had never smoked or if the respondent had quit smoking at least one month before participating in the study) and smoking (if the respondent smoked at least one cigarette/day, or if the last cigarette was smoked in the last month).

Participants were asked to report how often they consumed alcohol in the last year before the study. In addition, they were asked how often they consumed 1–2 standard doses of alcohol (SJA), assuming one dose equals 10 g of pure ethyl alcohol. From the possible answers, the respondent could choose from the following responses: I do not drink alcohol (in the case of not consuming alcohol completely), less than once a month, between once a month and once a week, and more than once a week.

### 2.3. Statistical Analysis

Categorical data are presented as absolute and relative frequencies, and numerical data as mean with standard deviation (SD). Comparisons were performed using Pearson’s Chi-square tests for categorical variables; numerical were analysed using the *t* test. Multivariable binary logistic regression was used to find the significant predictors of PHQ-9 ≥ 10. The backward elimination method was used (*p*-value < 0.01). Moreover, the ordinal logistic regression was performed to find the significant predictors associated with the odds of a higher category of PHQ-9. The assumption of proportional odds was tested using a full likelihood ratio test, comparing the fitted location model to a model with varying location parameters. *p* values < 0.05 was considered statistically significant.All statistical analyses were performed using IBM Corp. (released in 2019) and IBM SPSS Statistics for Windows, Version 26.0. (IBM Corp, Armonk, NY, USA).

## 3. Results

### 3.1. General Characteristics of Participants

Table 1 presents the characteristics of the study group in terms of gender. Out of 4040 people surveyed under the project, 3752 respondents were included in the research. More than half of the study group were women (58.66%, *n* = 2201), inhabitants of rural areas (66.9%, *n* = 2509) and married (88%, *n* = 3300). The mean age in the study group was 52 ± 8.1, while the women were slightly older than the men, but the age did not significantly differentiate the group (*p* = 0.19). Compared to men, women were better educated, smoked cigarettes, consumed alcohol less frequently, and had a normal BMI more often.

In the screening for depression symptoms using the PHQ-9 questionnaire in the entire population we studied, the risk of depression symptoms was 16.1% (*n* = 605), of which women (6.9 ± 3.5) obtained a significantly higher mean score than men (5.8 ± 3.4) *p* < 0.001. Moreover, when assessed by the PHQ-9 questionnaire, women more often than men obtained a point value greater than or equal to 10 (*p* < 0.001).

### 3.2. Relationship between Sociodemographic Variables and the Risk of Depression Symptoms

Table 2 shows the relationship between socioeconomic and health characteristics and the risk of developing symptoms of depression (PHQ-9 ≥ 10). Place of residence, education, living alone, BMI value and comorbidities (hypertension and/or diabetes and/or hypercholesterolaemia) were significantly associated with the risk of depression symptoms in women. A higher prevalence of symptoms of depression was observed in the study participants living in rural areas, with only primary education, living alone, with obesity and chronic diseases. Among men, marital status, education, living alone and having chronic diseases were associated with depression symptoms (PHQ-9 ≥ 10). A higher prevalence of depression symptoms was observed in single men (widowers or single men) with only primary education, living alone and suffering from chronic diseases. Taking into account age, both women and men who obtained 10 or more points in the PHQ-9 questionnaire were older compared to the respondents who obtained less than 10 points (women: 53 ± 7.6 vs. 51 ± 8.3; men: 54 ± 7.9 vs. 52 ± 8).

When the four categories of the PHQ-9 questionnaire were considered, in the case of women, significant relationships were obtained in precisely the same characteristics (age, place of residence, education, living alone, BMI value and coexisting diseases). However, in men, it was additionally observed that with the increase in the frequency of alcohol consumption, the percentage of people with symptoms of depression increases. Detailed analyses are presented in the Appendix A.

### 3.3. Relationship between the Risk of Developing Depression Symptoms and Socioeconomic and Health Variables in Multivariate Models in Gender Strata

Figure 2 shows significant predictors of PHQ-9 ≥ 10 by gender. In the case of women, living in rural areas, living alone, and having comorbidities significantly increased the chance of obtainingat least 10 pointsin the PHQ-9 questionnaire, while having at least a secondary education was associated with the re-education of the chances of achieving such a result. On the other hand, in men, the chance of PHQ-9 ≥ 10 was significantly increased by: living alone, smoking, and having comorbidities. Furthermore, graduating from at least a vocational school was associated with a decrease in the chance of havinga PHQ-9 ≥ 10.

Table 3 presents the results of ordinal logistic regression, presented as ORs, which indicate the probability of transition to the higher category according to the PHQ-9. Among women, apart from the features which were significant predictors of obtaining PHQ-9 ≥ 10, i.e., living in rural areas, education, living alone and having comorbidities, the categories of BMI assessment turned out to be significantly associated with obtaining a higher score in the PHQ-9 questionnaire. For obese women, the chance of being in the higher PHQ-9 category was 1.41 times higher than women with normal body weight. In men, apart from education, living alone and smoking, age and the declared frequency of alcohol consumption were significantly associated with obtaining a higher score in the PHQ-9.Increasing the age by one year increased the chance of being in a higher category by 1.02 times. In addition, the chance of being in a higher category, as assessed by the PHQ-9 questionnaire, among men who drink alcohol more than once a week was 1.7 times higher compared to men who do not drink or drink alcohol occasionally.

## 4. Discussion

Depression is one of the leading causes of disability worldwide and contributes to the deterioration of functioning and quality of life on many levels, including at the socioeconomic level of those impacted [28]. Many factors influence the appearance of symptoms of depression, including social, cultural, psychological, and biological factors. In this cross-sectional study, we estimated the prevalence of depression symptoms in a community with high deprivation rates and assessed how specific socioeconomic and health predictors affect the prevalence of depression symptoms by gender. In summary, our study’s results indicate that the percentage of respondents who obtained 10 or more points in the PHQ-9 questionnaire was over 16%. In addition, it was observed that women had significantly higher mean scores as assessed by the PHQ-9 questionnaire and obtained a point value of 10 and more than men significantly more often. Furthermore, our study confirms and highlights previous findings regarding the relationship between PHQ-9 and SES. While this finding is not surprising, it emphasises that socioeconomic determinants require more attention when screening for depression, especially in local communities with low SES.

As shown in the Greek [29], German [30], Peruvian [31], United States [32], or Russia [33] populations, and in our study, women suffered from depression more often than men. This result was confirmed in a study covering 24 European countries, with the most significant differences observed in Southern Europe, the former Soviet Union and Poland [34]. While differences in the severity of depression symptoms between the sexes indicate the involvement of genetic, neurohormonal and/or psychobiological factors, country differences in the studies cited above indicate that social conditions also play an important role in determining the prevalence of depression in the gender strata. It can also be assumed that the differences in depression symptomatology between women and men may result from the greater tendency of women to be more open to sharing their psychological problems with others [35]. Moreover, prejudices related to the stigma of depression which functions in society, may be less important for women than men [36]. Therefore, when completing the PHQ-9 questionnaire, women may have been less reluctant to admit that they had certain symptoms. On the other hand, gender differences in the incidence of depression may be related to biopsychosocial factors (such as hormonal differences between men and women resulting from the first menstruation, pregnancy, menopause and use of contraceptives), factors related to the family environment, childhood stressors and socio-cultural factors. Moreover, socio-cultural factors such as stress related to traditional roles of women may contribute to the higher incidence of depression [37].

The relationship between depressive symptomatology and age varies across studies. Generalising the research results available in the literature, the relationship between age and depression was much more significant in high-income countries compared to low and middle-income countries. In high-income countries, the risk of developing depression was associated with a young age. In contrast, in low and middle-income countries, the incidence of depression was higher in the elderly [38]. We made interesting observations in our research. Namely, the results of our research indicate that in the case of men, an increase in age by one year was associated with an increase in the chance of being in a higher category in the assessment of the PHQ-9 questionnaire, 1.02 times. However, there was no such relationship among women. In the studies of Hsu et al. [39] carried out in a group of 8422 people divided into persons performing physical exercise and not performing physical exercise, in the entire study population, it was observed that the level of depression was negatively correlated with age. On the other hand, the results of the research conducted on the Korean cohort of 4949 people aged 19–95 showed that people aged ≥70 years had the highest incidence of depression according to the PHQ-9 questionnaire, amounting to 11.2%. In contrast, the highest incidence of depression was observed in the age groups 19–29 years, 8.2%. Wang et al. [40] found that depressive symptoms peaked in people aged 30–40 and 80–90, which is consistent with the distribution of U-shaped depression in age groups. On the other hand, Jorm [41] found no consistent pattern in studies on the prevalence of depression depending on age groups. additionally, the conclusion from the research by Yang [42] is that ageing itself is not necessarily associated with the intensification of symptoms of depression. Young people may be prone to stressful events such as looking for a job, getting married, or the desire to achieve economic independence [43], while older people may have an increased risk of depression due to factors such as, inter alia, comorbidities, loss of a spouse, unemployment, reduced functional capacity or other social problems [44]. Therefore, we believe that understanding the factors which influence the prevalence of depression in different generations in particular countries or local communities is probably more important than estimating its prevalence by age. Our research results indicate that such searches should be conducted, especially among men.

Studies on the prevalence of depression symptoms among urban and rural residents show conflicting results. In our research, we observed that women living in rural areas significantly more often scored 10 points or more in the PHQ-9 questionnaire, while this correlation was not observed among men. Contrary to the results of our research, Michas et al. [29] observed a greater intensity of depression symptoms in large urban areas of Greece (Attiki and Central Macedonia) compared to less populated areas such as Crete and other Greek islands or Mainland. Romans et al. [45] analysed the Canadian Community Health Survey 1.2 (CCHS 1.2) data and found that the incidence rate of depression was significantly higher among urban residents. In the cited studies, the authors emphasize that urban and rural environments are not the same. This approach partly explains our result, which is different from the one described earlier. Rural borders, located near large cities differ from truly rural areas in that their inhabitants have a higher percentage of people with higher education and high income. They are more often married and have a higher level of health. Interesting conclusions in their research on the occurrence of lower levels of depression symptoms among city dwellers were published by scientists from the University of Chicago [46]. They proposed a model of depression driven by an individual’s accumulated experience mediated by social networks, which they validated for US cities using four independent datasets. The authors suggest that larger urban environments and urbanisation can provide greater social stimulation and socioeconomic linkages, which can act as buffers against depression. This theory is inconsistent with other research results [47,48], as mentioned in a letter to the editor by Huth et al. [49]; however, it may partially explain the obtained results of our own research, different from that of other authors. Other interesting theories explain the occurrence of depression through exposure to artificial light at night (ALAN) [50] and air pollution [51], with these factors being more pronounced in cities. Nevertheless, our research emphasizes the need to understand the high prevalence and innumerable determinants of depression among women living in the countryside.

In most studies, marital status and living with family members are viewed as factors associated with the incidence of depression [31,52,53]. Divorced people, widows/widowers, people who have not been married and living alone are at risk of developing depression. Bromet et al. [8] found that the demographic factor most strongly correlated with depression in high-income countries to being separated from a partner; in contrast, it was being divorced and widowed in low and middle-income countries. The results of our research confirmed that marital status significantly influenced the occurrence of symptoms of depression in the PHQ-9 assessment in the case of men, while living alone significantly increased the risk of depression symptoms in women and men. This fact can be explained by the specificity of the studied group, namely the local community with low SES; men are assigned the task of earning money to support the family as it is easier for them to find a job (especially in places with a high unemployment rate), and the salary they receive is higher. In addition, married men receive greater support from their partners, which is associated with a lower risk of depressive symptoms. Similarly, people who live alone do not have the support of a spouse or life partner. However, we could not confirm this in our research because we did not study the social support respondents receive.

In the authors’ study, the level of education was also a factor associated with depression symptoms in both women and men. The highest risk of depression symptoms was reported among people who only graduated from primary school; what is more, the increase in the level of education was associated with a decrease in the risk of depression symptoms. The result obtained by us is consistent with the results of other studies showing that people with higher education have fewer symptoms of depression [38,54]. It is assumed that the level of education positively impacts mental health, as it leads to better self-healing and active solving of stress problems [55]. Moreover, people with higher education have lower financial burdens and better working conditions, which may act as an intermediary in the relationship between symptoms of depression and education [56].

Several cross-sectional studies and prospective studies have shown a strong association between cigarette smoking and the risk of developing symptoms of depression [29,54], which has been confirmed in our study in men. When it comes to alcohol consumption, there is a two-way relationship with the occurrence of symptoms of depression [57]. Excessive consumption of alcohol can harm work and relationships and can lead to depression. On the other hand, alcohol abuse may be associated with trying to deal with symptoms of depression. Several studies have found that alcohol consumption was significantly associated with the occurrence of depression symptoms [54]. In this study, we found that men who reported drinking alcohol more than once a week were 1.7 times more likely to be classified in a higher category on the PHQ-9 questionnaire than men who did not drink or drank alcohol occasionally. We did not find any relationship between the occurrence of depression symptoms and alcohol consumption in women. Carpena et al. [58] observed results different from ours. In their research, they stated that it is among women that excessive alcohol consumption influences the incidence of depression symptoms. However, they did not observe this among men, who in Brazil consume alcohol more often than women.

A meta-analysis involving eight cross-sectional studies showed that the probability of symptoms is 35% higher in obese people compared to non-obese people [59]. Mulugate et al. [60], in their studies conducted on an English cohort, showed that symptoms of depression are associated with concomitant obesity in both women and men. On the other hand, another meta-analysis involving 15 studies showed a 38% higher probability of developing symptoms of depression in people with symptoms of central obesity, with higher estimates observed in women compared with men [61]. In our research, it was observed that obese women more often reported the occurrence of depression symptoms and we observed that for obese women, the chances of being in a higher category according to the PHQ-9 questionnaire were 1.41 times higher compared to women with normal body weight. The mechanism underlying the relationship between obesity and depression remains unclear. However, molecular and clinical studies provide some evidence for the involvement of the hypothalamic-pituitary-adrenal axis (HPA), an increase in some markers of inflammation and insulin sensitivity [62].

Excessive body weight impacts the deregulation of the HPA axis, elevation of some inflammatory markers and the development of insulin resistance, which is observed in both obese and depressed people [63]. In addition, the above-mentioned disorders may impact the secretion or metabolism of neurotransmitters such as serotonin, norepinephrine and dopamine in the brain and, consequently, impact mood [64]. In the face of the growing obesity epidemic, tackling its causes and consequences remains a key public health priority. Furthermore, as our research shows, obesity seems to be an important risk factor for the development of symptoms of depression in women, who should constitute an important target group for the implementation of prophylactic strategies for both depression symptoms and obesity-related lifestyle.

In our study, comorbidities were significantly associated with an increase in the PHQ-9 questionnaire in both women and men and thus with the severity of depression symptoms. Similar findings among patients with multiple sclerosis were obtained in cross-sectional studies from the USA [65], in patients with lupus erythematosus [66] or patients with type 2 diabetes [67]. While Chin et al. [68] found that having at least two chronic diseases increases the risk of depression symptoms. Depression in patients with chronic diseases may be related to many factors, including higher mortality or poor prognosis of the disease [69]. Moreover, depression is associated with an approximately 50% increase in the cost of caring for a person with a chronic disease [70]. Therefore, detection and counselling in the field of coping with symptoms of depression in chronically ill patients can improve treatment outcomes and reduce treatment costs. Whether individuals adapt well to their chronic illness and remain mentally healthy depends on many different aspects of the individual and their social environment. Understanding the factors in which symptoms arise and, conversely, the buffers which prevent the onset of depression symptoms, i.e., in which conditions patients adapt well to the condition associated with a chronic disease, can help ensure appropriate, targeted treatment and prevention.

### The Strengths and Limitations

The strengths and weaknesses of this study deserve consideration. First, we used the PHQ-9 questionnaire, characterised by good reliability, to assess the symptoms of depression [26]. Secondly, it is a study conducted on a relatively large number of respondents in the local community, with one of the highest mortality rates in Poland in 2009–2011 due to cardiovascular diseases. Third, the sample size allows us to better understand the specificity of the occurrence of depression symptoms, especially in their multifactorial conditions. Fourth, the research was carried out in a local community characterized by a high deprivation rate compared to the population of the region and the country in terms of low income, high unemployment, low living conditions, low education, and more limited access to goods. Fifthly, our research results identified the socioeconomic and health risk factors associated with SES, which influence the occurrence of depression symptoms. A better understanding of the risk factors associated with symptoms of depression can have significant public health benefits.

Our study also has several limitations which must be taken into account. Firstly, the respondents participating in the research came from south-eastern Poland, from one county; therefore, the results cannot be generalised to match the entire population. Secondly, the cross-sectional design of this study and the conducted analysis of the results limits its power to cause-and-effect inference, as it only shows a certain tendency of relationships. Third, although we assessed many SES-related variables in the study, the impact of residua or hidden confounding variables cannot be excluded, as is the case with all cross-sectional studies. Fourth, the PHQ-9 questionnaire is used for screening, not depression diagnosis. Therefore, the incidence of depression symptoms in our research may have been overestimated.

## 5. Conclusions

Summarising the results of studies conducted in a local community characterized by a high deprivation rate, socioeconomic and health variables related to SES significantly impact the incidence of depression, but they differ in terms of gender. To combat the growing burden of depression in local communities, screening is needed in primary care settings to detect the first symptoms of depression. In local communities with high levels of deprivation, screening should, in particular, include: older adults living in rural areas, less educated, suffering from chronic diseases, living alone or overweight. Of course, screening should be coupled with publicly funded interventions to increase the SES of those involved. The research material described in this manuscript was collected from a local community with a high deprivation rate, but also the community characterized by a high mortality rate due to cardiovascular diseases. Effective prevention of CVC disease at the individual level should also be based on the diagnosis of patients with mental disorders, as such patients have many more lifestyle-related risk factors that need to be recognized and treated. In addition, mental health care improves the quality of life, reduces symptoms of stress and the risk of suicide, and may improve cardiovascular health [71]. Our finding is valuable in light of insufficient research on the role of socioeconomic and health factors in the risk of depression symptoms. However, longitudinal studies are needed to confirm our results.

## Figures and Tables

**Figure 1 ijerph-19-11797-f001:**
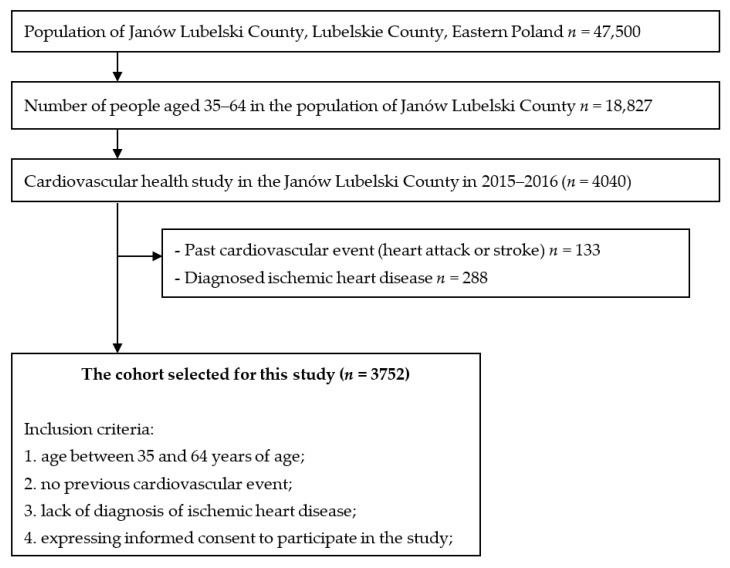
Study design.

**Figure 2 ijerph-19-11797-f002:**
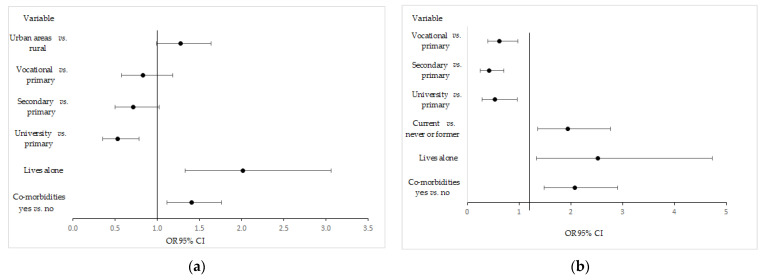
The significant predictors of getting the PHQ-9 ≥ 10 in women (**a**) and men (**b**).

**Table 1 ijerph-19-11797-t001:** Characteristics of the researched group according to their gender.

Variables	Female (*n* = 2201)	Male (*n* = 1551)	Total (*n* = 3752)	*p*
Age [years]	52 ± 8.2	52 ± 8.0	52 ± 8.1	0.19
Rural areas	1438 (65.3)	1071 (69.1)	2509 (66.9)	0.02
Marital status:				
Married	1917 (87.1)	1383 (89.2)	3300 (88.0)	<0.001
Single (bachelor/bachelorette)	125 (5.7)	147 (9.5)	272 (7.2)	
Widow/widower	159 (7.2)	21 (1.4)	180 (4.8)	
Education:				
Primary	223 (10.1)	190 (12.3)	413 (11.0)	<0.001
Vocation	667 (30.3)	723 (46.6)	1390 (37.0)	
Secondary	776 (35.3)	428 (27.6)	1204 (32.1)	
University	535 (24.3)	210 (13.5)	745 (19.9)	
Smoking status:				
Yes	250 (11.4)	345 (22.2)	595 (15.9)	<0.001
No	1951 (88.6)	1206 (77.8)	3157 (84.1)	
Alcohol consumption:				
No or less than once a month	2140 (97.2)	1205 (77.7)	3345 (89.2)	<0.001
Between once a month and once a week	42 (1.9)	195 (12.6)	237 (6.3)	
More than once a week	19 (0.9)	151 (9.7)	170 (4.5)	
Lives alone	116 (5.3)	58 (3.7)	174 (6.64)	0.03
BMI [kg/m^2^]:				
Normal [18.5–24.99 kg/m^2^]	633 (28.9)	272 (17.6)	905 (24.2)	<0.001
Overweight [25–29.99 kg/m^2^]	794 (36.3)	716 (46.3)	1510 (40.4)	
Obesity [≥30 kg/m^2^]	762 (34.8)	560 (36.2)	1322 (35.4)	
Comorbidities (Yes) ^#^	661 (30.0)	427 (27.5)	1088 (29)	0.09
Patient Health Questionnaire (PHQ-9):				
Total score	6.9 ± 3.5	5.8 ± 3.4	6.4 ± 3.5	<0.001
None (0–4)	574 (26.1)	579 (37.5)	1153 (30.7)	<0.01
Mild (5–9)	1196 (54.3)	798 (51.5)	1994 (53.1)	
Moderate (10–14)	366 (16.6)	150 (9.7)	516 (13.8)	
Moderately-Severe (15–19)	19 (1.2)	51 (2.3)	70 (1.9)	
Severe (20–27)	5 (0.36)	14 (0.64)	19 (0.55)	
PHQ-9 (≥10)	431 (19.6)	174 (11.2)	605 (16.1)	<0.001

^#^ Comorbidities: hypertension and/or diabetes and/or hypercholesterolemia.

**Table 2 ijerph-19-11797-t002:** Relationship between socioeconomic and health variables and the risk of depression symptoms in the study group.

Variables	Patient Health Questionnaire (PHQ-9)					
	Female (*n* = 2201)		*p*	Male (*n* = 1551)		*p*
	0–9	≥10		0–9	≥10	
Age [years]:	51 ± 8.3	53 ± 7.6	<0.001	52 ± 8	54 ± 7.9	0.002
Place of living:						
Rural areas	1133 (78.8)	305 (21.2)	0.008	947 (88.4%)	124 (11.6)	0.5
Urban areas	637 (83.5)	126 (16.5)		430 (89.6%)	50 (10.4)	
Marital status:						
Married	1550 (80.9)	367 (19.1)	0.38	1239 (89.6%)	144 (10.4)	0.03
Single (bachelor/bachelorette)	98 (78.4)	27 (21.6)		121 (82.3%)	26 (17.7)	
Widow/widower	122 (76.7)	37 (23.3)		17 (81%)	4 (19)	
Education:						
Primary	162 (72.6)	61 (27.4)	<0.001	155 (81.6)	35 (18.4)	0.002
Vocation	522 (78.3)	145 (21.7)		638 (88.2)	85 (11.8)	
Secondary	626 (80.7)	150 (19.3)		394 (92.1)	34 (7.9)	
University	460 (86)	75 (14)		190 (90.5)	20 (9.5)	
Smoking status:						
Yes	194 (77.6)	56 (22.4)	0.23	287 (83.2)	58 (16.8)	<0.001
No	1576 (80.8)	375 (19.2)		1090 (90.4)	116 (9.63)	
Alcohol consumption:						
No or less than once a month	1718 (80.3)	422 (19.7)	0.21	1075 (89.2)	130 (10.8)	0.16
Between once a month and once a week	38 (90.5)	4 (9.5)		175 (89.7)	20 (10.3)	
More than once a week	14 (73.7)	5 (26.3)		127 (84.1)	24 (15.9)	
Lives alone						
Yes	79 (68.1)	37 (31.9)	0.001	43 (74.1)	15 (25.9)	0.001
No	1691 (81.1)	394 (18.9)		1334 (89.4)	159 (10.6)	
BMI [kg/m^2^]:						
Norm [18.5–24.99 kg/m^2^]	536 (84.7)	97 (15.3)	<0.001	240 (88.2)	32 (11.8)	0.86
Overweight [25–29.99 kg/m^2^]	641 (80.7)	153 (19.3)		639 (89.2)	77 (10.8)	
Obese [≥30 kg/m^2^]	584 (76.6)	178 (23.4)		495 (88.4)	65 (11.6)	
Comorbidities ^#^						
Yes	500 (75.6)	161 (24.4)	<0.001	355 (83.1)	72 (16.9)	<0.001
No	1270 (82.5)	270 (17.5)		1022 (90.9)	102 (9.1)	

^#^ Comorbidities: hypertension and/or diabetes and/or hypercholesterolemia.

**Table 3 ijerph-19-11797-t003:** Relationship between selected socioeconomic and health variables and the risk of depression in gender strata.

Variables	OR	95% CI	*p*
Female:			
Place of living:			
Urban areas	1		
Rural areas	1.206	1.005–1.448	0.044
Education:			
Primary	1		
Vocation	0.982	0.730–1.321	0.903
Secondary	1.058	0.787–1.422	0.710
University	0.691	0.505–0.944	0.02
Lives alone:			
No	1		
Yes	1.731	1.204–2.448	0.003
BMI [kg/m^2^]:			
Norm [18.5–24.99 kg/m^2^]	1		
Overweight [25–29.99 kg/m^2^]	1.200	0.977–1.474	0.081
Obesity [≥30 kg/m^2^]	1.407	1.127–1.756	0.003
Comorbidities: ^#^			
No	1		
Yes	1.264	1.046–1.529	0.015
Male:			
Age:	1.022	1.035–1.035	0.001
Education:			
Primary	1		
Vocation	0.804	0.588–1.099	0.172
Secondary	0.658	0.470–0.923	0.015
University	0.747	0.503–1.109	0.148
Smoking status:			
No	1		
Yes	1.546	1.218–1.961	<0.001
Alcohol consumption:			
No or less than once a month	1		
Between once a month and once a week	1.296	0.966–1.740	0.084
More than once a week	1.656	1.187–2.311	0.003
Comorbidities: ^#^			
No	1		
Yes	1.431	1.42–1.794	0.002

^#^ Comorbidities: hypertension and/or diabetes and/or hypercholesterolemia.

## Data Availability

The datasets used and/or analyzed during the current study are available from the corresponding author on reasonable request.

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
