# Peer review of "The Prevalence of Depression Symptoms and Their Socioeconomic and Health Predictors in a Local Community with a High Deprivation: A Cross-Sectional Studies"

_ijerph, 2022, doi:10.3390/ijerph191811797_

Round 1

Reviewer 1 Report

The survey is extremly important and results of depression based on the PHQ-9 questionnaire, which is reliable and used in many countries what makes the conclussions to be generated. The study was conducted on a relatively large number of participants however the respondants lived in the local community in one of the province of Poland. Althought Poland might be not highly representative country for the epidemiology of depression, the region of research was one of the highest mortality rates in Poland due to cardiovascular diseases. This fact is highly important for better understanding of the specificity of the occurrence of depression symptoms, especially in their multifactorial conditions related to SES (low income, high unemployment, low living conditions, low education, and more limited access to goods).

Authors identified the socioeconomic and health risk factors associated with SES which influence the occurrence of depression symptoms.

Conclussions are well presented and lead the readers to better understanding of the risk factors associated with depression symptoms. The results are significant for improving the system of public health not only in local context but international as well.

Only one explanation should be add why the inclusion criterion was  the age between 35 and 64 years of age. Young people (<35 years) are also very vulnerable to depression.

the article is written in a very communicative way.

Reviewer 2 Report

As a sociologist, my expertise to revise some specific more technical aspects of the paper, especially those related to the test used is limited.

However, I have two major comments for the authors. The first is that the scientific quality of the study seems robust. The second is that the provenience of the sample from southern Poland seems to limit the representativeness of the paper on a larger scale, as the authors themselves have highlighted. But there it comes my observation, the relevance of this paper lay not in its replicability but in the relationship between socioeconomic dynamics and depression, a finding that seems to counter some mainstream literature that presents us with the socio-biological argument.  Published as it is, this is a good paper but an average one. Re-elaborated to improve the research design to explain the causes (migration, economic crisis) of the depression in those particular groups would likely make of it a great paper. How not to have researched if the women were also victims of house violence for instance?

Good luck with the paper.

Reviewer 3 Report

A very well conducted study. The findings will be of relevance to public health administrators, medical professions, and as well the data will add to the broader body of knowledge on the subject.

The research question is clear, the literature review is thorough and the methodology is appropriate to the study.

The data analysis and discussion supports the findings and conclusions.

Reviewer 4 Report

This paper tackles an interesting question on deprivation and mental health but could benefit from these improvements. 

1. Consider using theory in a paragraph or so. Deprivation theory works or something on the social determinants of mental health. 

2. Could the deficits addressed here be complemented by assets in the community such as religious involvement or social capital?

3. Make a stronger case for study significance. Why this county versus others that are similar?

4. Provide greater treatment of study implications. 

5. Shorten the title. It is far too long. 

6. Copy editing would be welcome. 

Round 2

Reviewer 4 Report

I commend the authors on their revision. The paper could still use very careful proofreading. It seems that there is even a typographical error in the paper title.